# Traditional, Complementary and Alternative Medicines in the Treatment of Ejaculatory Disorders: A Systematic Review

**DOI:** 10.3390/medicina59091607

**Published:** 2023-09-06

**Authors:** Kristian Leisegang, Chinyerum Sylvia Opuwari, Faith Moichela, Renata Finelli

**Affiliations:** 1School of Natural Medicine, University of the Western Cape, Bellville 7535, South Africa; kleisegang@uwc.ac.za; 2Department of Medical Biosciences, University of the Western Cape, Bellville 7535, South Africa; copuwari@uwc.ac.za (C.S.O.); 4218138@myuwc.ac.za (F.M.); 3Create Fertility, London EC2V 6ET, UK

**Keywords:** ejaculatory dysfunction, ejaculation, traditional medicine, complementary medicine, alternative medicine

## Abstract

*Background and Objectives*: Ejaculatory dysfunction (EjD) is a common male sexual disorder that includes premature ejaculation, delayed ejaculation, retrograde ejaculation, and anejaculation. Although psychological and pharmacological treatments are available, traditional, complementary, and alternative medicine (TCAM) is reportedly used. However, the clinical evidence for TCAM in EjD remains unclear. Therefore, this study aims to systematically review human clinical trials investigating the use of TCAM to treat EjD. *Materials and Methods*: A systematic review of the literature following the Preferred Reporting Items for Systematic Reviews and Meta-Analyses (PRISMA) guidelines was conducted by searching Scopus and PubMed databases. Controlled clinical trials investigating a cohort of male patients diagnosed primarily with EjD and undergoing any TCAM intervention compared to any comparison group were included. Quality of the studies was assessed using the Cochrane Risk of Bias tool for randomized controlled trials. *Results*: Following article screening, 22 articles were included. Of these, 21 investigated TCAM in premature ejaculation, and only 1 investigated TCAM in retrograde ejaculation. Different TCAM categories included studies that investigated lifestyle, exercise and/or physical activities (n = 7); herbal medicine supplements (n = 5); topical herbal applications (n = 4); acupuncture or electroacupuncture (n = 3); vitamin, mineral and/or nutraceutical supplements (n = 1); hyaluronic acid penile injection (n = 1); and music therapy (n = 1). Only 31.8% (n = 7) of the included studies were found to have a low risk of bias. The available studies were widely heterogenous in the TCAM intervention investigated and comparison groups used. However, the included studies generally showed improved outcomes intra-group and when compared to placebo. *Conclusions*: Different TCAM interventions may have an important role particularly in the management of PE. However, more studies using standardized interventions are needed.

## 1. Introduction

Ejaculatory dysfunction (EjD) is a multidimensional psych-neuro-uro-endocrine pathological condition characterized by the disruption of the ejaculatory cascade that results in a failure of normal ejaculation [1]. Sub-categories commonly include expulsion phase disorders such as premature ejaculation (PE), delayed ejaculation (DE) and anejaculation; emission phase disorders such as retrograde ejaculation (RE); and orgasm disorders such as anorgasmia [2]. Due to the overlap of neuromuscular control, these disorders are grouped as EjD but may have a different etiology, pathology, diagnosis and management [2].

Data on the epidemiology in the general population of EjD as a heterogenous sexual disorder are scant [2]. The most common form of EjD in the general population is PE, affecting up to 20–30% of the male adult population [2,3], where RE is considered the second most common type of EjD and an important cause of infertility [4]. Other types of EjD such as DE and anejaculation are estimated to affect 1–4% of sexually active men [5]. However, EjD is reported in 1.8% of males with infertility, with RE the most common (0.7% males), followed by PE (0.5%) and anejaculation (0.4%) [6].

Lifelong PE has been defined as ejaculation that nearly always occurs within or around 1 min following vaginal penetration since the first sexual experience, whereas acquired PE is a significant reduction in latency time of 3 min or less [7]. The intravaginal ejaculation latency time (IELT/ELT) is widely used in clinical trials to assess ejaculation delay in men with PE [8]. IELT/ELT is recommended to be performed through stopwatch measurement from vaginal intromission to the start of intravaginal ejaculation [8]. In the general population, the median (range) IELT is 5.4 (0.55–44.1) min, with a significant decrease for 51 years and older compared to 18–30 years of age [8]. In addition to the IELT, numerous questionnaires have been validated for PE. These include the Premature Ejaculation Profile (PEP) [9], Premature Ejaculation Diagnostic Tool (PEDT) [10], Arabic Index of Premature Ejaculation (AIPE) [11], Chinese Index of Premature Ejaculation-5 (CIPE-5) [12], Male Sexual Health Questionnaire-Ejaculatory Disorder (MSHQ-EjD) [13], Checklist for Early Ejaculation Symptoms (CHEES) [14], Index of Premature Ejaculation (IPE) [15] and Multiple Indicators of Premature Ejaculation [14]. RE is due to the lack of closure of the internal urethral sphincter (bladder neck) that results in the reflux of semen into the bladder (hypospermia or aspermia) [16], and typically presents with absent or intermittent seminal emission with orgasm and spermatozoa identified in post-coital urine samples [17]. DE and anorgasmia are described in men with a delay or absence of orgasm that is ongoing or recurrent [18]. Although no formal cut-off is defined, DE is proposed to occur with ejaculation latencies of >20 or 30 min [5,19]. 

Selective serotonin reuptake inhibitors (SSRIs), tricyclic antidepressants, phosphodiesterase type 5 (PDE5)-inhibitors and/or topical anesthetics are recommended in the management of primary PE. In contrast, acquired PE management is typically dependent on the underlying etiology [3,20]. SSRIs, such as dapoxetine and paroxetine, have strong clinical evidence for delaying ejaculation in acquired and lifelong PE [21]. Management of RE is primarily aimed at the underlying cause, and alpha-agonists have been used off-label without supporting evidence [20]. Pharmacological treatments for DE, anejaculation and anorgasmia, however, remain limited, with few large-scale studies showing limited efficacy and/or significant adverse events [22], although psychotherapy and addressing underlying causes are recommended [18]. In addition, traditional, complementary and alternative medicine (TCAM) is widely used for various male sexual dysfunctions in the general population, yet is poorly investigated for clinical efficacy [23,24,25,26]. 

Traditional knowledge reflects the interaction between individuals and societies with nature, and has the potential to contribute significantly to the healthcare of local people, as well as to scientific knowledge and drug discovery through documentation of traditional medicine and local ethnobotany [27,28,29]. As broadly reported by the WHO [30] and Cochrane collaboration [31], TCAM incorporates numerous treatment modalities, including nutritional medicine; dietary supplements; lifestyle changes; exercise therapies; herbal medicines; traditional Chinese medicine and acupuncture; physical therapies such as massage, cupping and acupressure; homeopathy; music therapy; breathing therapy; energy medicine (such as reiki); mind–body techniques (such as meditation and yoga); and psychotherapy and counselling [30,31,32]. Although numerous TCAM approaches are used for male sexual function, there are few human clinical studies on ejaculatory disorders and TCAM [23,24,25,26]. Therefore, this study aims to systematically review human clinical trials investigating the use of TCAM to treat EjD using validated outcome measures.

## 2. Materials and Methods

A literature search was conducted on 22 September 2022 using Scopus and PubMed databases, with no time limit, and followed the Preferred Reporting Items for Systematic Reviews and Meta-Analyses (PRISMA) guidelines [33]. 

The keywords were selected to identify any publication investigating the use of traditional and natural medicines [(“Phyto*” OR “plant*” OR “polyherb*” OR “herb*” OR “medicine*” OR “CAM” OR “acupuncture” OR “acupressure” OR “moxibustion” OR “tuina” OR “qigong” OR “hydrotherapy” OR “homoeopat*” OR “homeopat*” OR “aromatherap*” OR “aromatic oil*” OR “nutraceutical*” OR “supplement*” OR “nutrition*” OR “diet*” OR “reflexology” OR “massage” OR “exercise” OR “unani” OR “ayurveda” OR “cupping” OR “naturopat*” OR “yoga” OR “reiki” OR “meditation” OR “chiropractic” OR “osteopathy” OR “thermal therap*” OR “light therap*” OR “music therap*”) in human male patients [(“human*” OR “man” OR “men” OR “patient*”)] diagnosed with an ejaculatory disorder [(“ejaculat*” OR “ anejaculat*”)]. The asterisk (*) was used to include any possible version of the term searched. No further limitations, restrictions or search filters were used.

The following inclusion criteria were used for eligibility of studies: (i) a cohort of male patients diagnosed with any recognized EjD; (ii) any sole TCAM intervention in at least one intervention group; (iii) controlled clinical trials with any comparison group as study type; (iv) any relevant outcome measures for EjD; (v) English, Spanish and Italian full-text articles. The following exclusion criteria were used for eligibility of studies: (i) no control group; and (ii) observational studies, review studies, case reports, conference proceedings, pre-prints, and any other grey literature. 

After searching the databases with keywords and removal of duplicates, a preliminary literature screening was performed based on the title and abstract by authors (C.S.O., F.M., R.F.), followed by full-text retrieval of the remaining articles for eligibility analysis based on the inclusion and exclusion criteria. Any disagreement was solved after discussing with the first author (K.L.). Data were collected using an Excel sheet and included the study design, a description of the patient and control cohorts, the dosage and duration of the intervention used, a report of the outcomes and any adverse effects. To simplify the results and discussion, included studies were presented grouped in macro-areas based on the type of intervention used: herbal medicine supplements; topical herbal applications; vitamin, mineral and nutraceutical supplements; acupuncture and electroacupuncture; lifestyle, exercise and/or physical therapy; penile injections; and music therapy. Each included study’s quality was evaluated using the Cochrane Risk of Bias tool for randomized controlled trials (RCTs) [34].

## 3. Results

A total of 1575 articles were retrieved from Scopus and PubMed databases (Figure 1). After removing duplicates (n = 142), the abstract and the title were examined, and 1388 articles were excluded. Forty-five full texts were assessed for eligibility, with 23 articles excluded, resulting in 22 articles included for qualitative analysis. Reasons for articles excluded at eligibility included no full-length clinical trial (n = 7) [35,36,37,38,39,40,41], no TCAM intervention (n = 5) [42,43,44,45,46], no comparison group (n = 6) [47,48,49,50,51,52], no ejaculatory disorder cohort under investigation (n = 3) [53,54,55], and combined treatment with conventional medicine (n = 2) [56,57].

The data extracted from the included studies are summarized in Table 1 based on the type of intervention provided. Proportionately, 31.8% (n = 7) investigated different lifestyle, exercise and/or physical activities [58,59,60,61,62,63,64]; 22.7% (n = 5) investigated herbal medicine supplements [65,66,67,68,69]; 18.2% (n = 4) investigated topical herbal applications [70,71,72,73]; 13.6% (n = 3) investigated acupuncture or electroacupuncture [74,75,76]; and 4.5% (n = 1) each investigated vitamin, mineral and/or nutraceutical supplements [77]; hyaluronic acid penile injection [78]; and music therapy [79]. 

Of the 22 studies included, 95.5% (n = 21) studies investigated the effect of TCAM treatments on patients with PE [58,59,60,61,62,63,64,65,66,67,68,69,70,71,72,73,74,75,76,78,79], with only 1 study investigating RE [77]. No studies were identified that investigated TCAM for treating DE, anejaculation or other EjDs. Of the 21 studies that investigated TCAM in PE, most used the IELT as a primary outcome measure [58,59,60,62,63,65,66,67,68,69,70,71,72,73,74,75,76,78,79]. Furthermore, 42.9% (n = 9) used the PEDT alongside IELT [59,62,66,68,73,74,76,79] or as a primary outcome [61], 9.5% (n = 2) of studies used the PEP alongside IELT [67] or as a primary outcome [61], and 9.5% (n = 2) used the CIPE-5 alongside IELT [69,70]. In contrast, individual studies used the AIPE alongside IELT [78], the MSHQ-EjD alongside IELT [66], and the CHEES [64] as primary outcome measures of PE. For the single included study investigating RE, the antegrade ejaculation recovery rate was used as the primary outcome, defined as >1 seminal ejaculation through the external urethral meatus over a 4-week treatment period to identify a successful treatment [77]. 

The studies included in this review used a variety of different comparison groups. A placebo comparison was used in 27.3% (n = 6) of studies [65,67,71,72,73,78], with 9.1% (n = 2) of studies comparing treatment to a cohort receiving no treatment [64,69]. The use of SSRIs as a comparison group included 22.7% (n = 5) of studies using dapoxetine [59,60,68,76,79] and 4.5% (n = 1) of studies each using paroxetine [74] and fluoxetine [58]. The tricyclic antidepressant amoxapine was used as a comparison group in 4.5% (n = 1) of studies and the only included study investigating RE [77]. The sham acupuncture group was used in two studies investigating acupuncture treatment [74,76], which was further compared to paroxetine [74] or dapoxetine [76]. Longdan Xiegan was used as a comparison group investigating electroacupuncture [75]. Minimal physical activity or sedentary lifestyle [59], sexual behavioral treatment [66] and desensitization therapy [70] were used as a comparison in 4.5% (n = 1) of studies each. The remaining comparison groups are difficult to classify. Mobile coaching application of physical and mental distancing exercises was compared to verbal and printed instructions for physical and mental distancing exercises [61], penile root masturbation (PRM) was compared to Kegel exercises [62], sphincter control training without a masturbation device (GWD) was compared to sphincter control training without a masturbation device (GWtD) [63], and vibrator-assisted start–stop exercises and mindfulness meditation (VSS+) was compared to vibrator-assisted start–stop exercises (VSS) [64].

Most studies and interventions investigated showed few adverse effects and good short-term safety profiles. However, reporting the adverse events were not consistently mentioned in all the studies included and was not reported at all in four studies [61,66,69,79] (Table 1). Only 31.8% (n = 7) of the included studies were found to have a low risk of bias [67,70,71,72,73,77,78]. An intermediate risk of bias was found in 13.6% (n = 3) of the included studies [65,66,68]. A high risk of bias was identified in 54.5% (n = 12) of studies [58,59,60,61,62,63,64,69,74,75,76,79]. The risk of bias is summarized in Figure 2.

## 4. Discussion

Although there is reportedly an extensive use of TCAM for the management of male sexual dysfunction, including EjD [23,24,25,26], the extent of the available clinical trials of TCMA for EjD has remained unclear. Following a PRISMA-guided systematic literature search, only a few controlled clinical trials (n = 22) were identified that investigated a wide range of different interventions that can be considered TCAM (Table 1). Furthermore, most (n = 21) of these studies investigated TCAM in PE only, which can be explained as PE is considered the most common and significant EjD affecting the general adult population [2]. These results further suggest that there is an under-representation of TCAM investigations in other clinically important forms of EjD, with only one study investigating the use of vitamin B12 in retrograde ejaculation, and no studies investigating other forms of EjD. 

No previous similar systematic review to investigate different TCAM interventions on EjD has been identified in the literature. Previously, a 2017 systematic review on the use of CAM in PE included 10 studies in the final analysis, predominantly consisting of herbal medicine and acupuncture [80]. This present systematic review included 4 of the 10 studies that were previously reported in 2017 [69,71,72,74], while we excluded 6 of these studies due to Chinese language full text (n = 2) [81,82], combination treatment with conventional medicine (n = 3) [83,84,85] and no inter-group comparison (n = 1) [86]. Our systematic review on TCAM for EjD expanded on the previous review in 2017 [80] by widening the definition of TCAM and including all types of EjD, which further identified three additional studies published prior to the 2017 review [80]. Specifically, this included herbal medicine [65], yoga [58] and physio-kinesitherapy, biofeedback therapy and electrical stimulation of the perineal floor [60]. 

### 4.1. Lifestyle, Exercise and/or Physical Therapy

Most TCAM systems have lifestyle, exercise and/or physical therapy components as part of patient management, although these practices may vary widely [32]. Physical inactivity is considered a risk factor for PE [20]. Only one study included in this review investigated moderate physical exercise (running > 30 min 5 days per week), which improved IELT and PEDT intra-group as effectively as dapoxetine and more effectively than intra-group improvements from minimal physical activity (walking < 30 min 5 days per week) [59].

The pelvic floor is important in male sexual function, particularly for erection and ejaculation control. Pelvic floor therapy may be useful for males with sexual dysfunction, including EjD such as PE, as it is suggested to contribute to the broader bio-neuro-musculo-psycho-social approach in the management of EjD [87]. Kegel exercises consist of regularly repeated pelvic floor muscle exercises, first developed in 1948 and used in urinary incontinence. Subsequently, electrical stimulation was developed to manage pelvic floor dysfunction, and written and audio perineal re-education techniques were developed. Together, these contribute to the management of urinary incontinence, sexual dysfunction and anorectal canal dysfunction [88]. One study included in this systematic review found that physio-kinesitherapy, biofeedback therapy and electrical stimulation of the perineal floor (PFM) as rehabilitation therapy improved IELT as did dapoxetine, although dapoxetine had a significantly better effect on IELT than PFM [60]. Another study found that penile root masturbation (PRM) and Kegel exercise increased IELT and decreased PEDT, where PRM was significantly better than Kegel exercises [62]. PRM is further supported in an earlier, small uncontrolled study of nine patients, where regular PRM was shown to increase IELT and decrease PEDT, suggesting that PRM has a short-term beneficial effect for PE [50]. In another study included in this systematic review, sphincter control training with a masturbation device (GWD) improved IELT similarly to sphincter control training with the addition of a masturbation device (GWtD) [63].

Yoga is an ancient Ayurvedic practice that has increased in popularity globally over the last two decades [89], associated with improved psychosocial attributes such as better coping, mindfulness and support [90]. Yoga is increasingly used for male sexual dysfunction, including PE [89]. Training in males undergoing a yoga camp for 12 weeks significantly improved sexual function, including libido, sexual satisfaction, sexual performance and confidence, partner satisfaction, erection, ejaculatory control and orgasm intensity based on the Male Sexual Quotient [91]. In this systematic review, one study reported that 12 weeks of yoga increased IELT, although fluoxetine was more effective than yoga with more adverse effects [58]. Proposed mechanisms for the benefit of yoga in PE are improved neuro-psycho-physiological functioning, hormonal regulation, reduced sympathetic and increased parasympathetic activity, increased serum serotonin levels, pelvic floor muscle strengthening, improved body awareness (and ejaculatory control), and management of underlying psychological or physical risk factors [92]. Furthermore, yoga has positive effects in relieving depression, anxiety and stress [93,94].

Psychotherapy and behavior modification are commonly used in the treatment of PE, although there is a lack of evidence-based trials [20]. One study in this systematic review investigated a combination of psychotherapy with physical exercises for strengthening the pelvic floor muscles and cognitive exercises for sexual failure distancing, which led to a decrease in PEDT and an increase in PEP [61]. There was a concomitant improvement in intercourse satisfaction and overall satisfaction, alongside decreased depression scores. Furthermore, this study found that the application of a mobile training app for exercises was more effective than written and verbal instructions [61]. One study investigated mindfulness meditation as a psycho-neuro technique for PE, where mindfulness meditation with vibrator-assisted start–stop exercises and (VSS+) improved CHEES significantly better compared to only vibrator-assisted start–stop exercises (VSS) and no treatment [64]. 

### 4.2. Herbal Medicine Supplements

Herbal medicines have been used for millennia to prevent and treat disease, with a large proportion of modern pharmaceuticals based on medicinal plant isolates [27,28,29,95,96]. Extractions of herbal medicines contain secondary plant metabolites that can work synergistically or as isolates on human physiology [95]. Herbal medicine is commonly used to enhance or improve male sexual function, with over 700 species used traditionally for this purpose [23,24]. *Hypericum perforatum* L. (Saint John’s wort) is used traditionally in Chinese, Greek and Islamic medicine systems and is a popular medicinal herb globally, prominently as an anti-depressant treatment [27,28,97]. This was the only herbal medicine investigated as a single herb with a standardized extract (hypericin). IELT increased significantly compared to the placebo control for *Hypericum perforatum* L., and improved intercourse satisfaction and orgasmic function [65]. In the treatment of mild, moderate and major depression, *Hypericum perforatum* L. is as effective as SSRIs, with lower side-effects and fewer trial dropouts compared to SSRI treatment groups [98,99], particularly through hypericin and hyperforin as well established active constituents [97]. 

OLNP-05 is an Ayurvedic polyherbal combination of *Mucuna pruriens* L., *Cynara cardunculus* L., *Trigonella foenum-graecum* L., *Withania somnifera* L. and L-arginine [67]. IELT increased significantly compared to placebo for OLNP-05, with an increased PEP and clinical global impression-improvement scale [67]. This formula is proposed to increase testosterone, dopamine, serotonin and nitric oxide (NO), and inhibit phosphodiesterase-5, although these mechanisms were not investigated in this study [67]. However, dopamine, serotonin and NO are critical mediators of ejaculatory control, and PDE5-inhibitors are commonly used in the pharmacological management of PE [20]. 

Gu-jing-mai-si-ha (GJMSHT) is a traditional Uighur medicine polyherbal combination of *Radix anacycli pyrethri*, *Mastiche*, *Fructus cardamomi*, *Rhizoma Cyperi*, *Stigma Croci*, *Semen Myristicae*, *Radix Curcumae*, *Folium Syringae oblatae*, *Radix et Rhizoma Nardostachyos*, *Fructus Tsaoko* and *Flos Rosae rugosae* [69]. GJMSHT increased IELT alongside CIPE10 and CIPE5 scores compared to a group with no treatment, with an increased Sexual Partner’s Satisfaction Rate and Wish Fulfillment Rate [69]. The increase may be mediated through NO and prostaglandin-F2α (PGF2α) [69]. NO levels are reduced in males with PE, where increased NO secondary to SSRI treatment improves ejaculation latency through a central mechanism [100]. Although the role of PGF2α in ejaculation is not clear, administration to boars significantly increased ejaculation time [101]. 

EiacuMev^®^ is a polyherbal combination of *Satureya montana*, *Tribulus terrestris*, *Phyllanthus emblica* and *Cardamomo*, with L-tryptophan, ascorbic acid, vitamins B1, B3 and B6 [66]. IELT increased significantly, and PEDT decreased significantly for EiacuMev^®^ compared to counselling and sexual behavioral treatment, with no change in MSHQ-EjD [66]. EiacuMev^®^ also increased FSH compared to counselling and sexual behavioral treatment, with no change for LH and testosterone [66]. Although the role of FSH, if any, in ejaculation has not been well investigated, men with PE have been found to have higher levels of FSH compared to healthy controls [102]. It is unclear if the increase in FSH with EiacuMev^®^ is beneficial for patients with EjD, which appears to be a somewhat contradictory result. *Satureja montana* is widely used in male sexual dysfunction and has been shown to delay ejaculation latency in rats [103]. *Tribulus terrestris* improves sexual behavior in rats through increased mount frequency and penile erection index; however, this also decreased ejaculatory latency and intromission latency [104]. 

Qiaoshao Formula (QSF) is a traditional Chinese herbal medicine composed of *Fructus Forsythiae*, *Radix Paeoniae Alba*, *Radix Bupleuri*, *Radix Astragali seu Hedysari*, *Morinda officinalis How*, *Rhizoma Dioscoreae* and *Rhizoma Acori Graminei* [68]. Although QSF improved IELT in an intra-group analysis, dapoxetine was significantly better than QSF [68]. However, the Chinese Medicine Symptoms Score decreased, and the sex life satisfaction score increased significantly more than dapoxetine alone [68]. In a separate excluded study, the combination of QSF and dapoxetine improved IELT in a sub-group of men (baseline IELT < 1 min and age < 30 years) significantly better than dapoxetine alone [57]. Furthermore, QSF has been shown to increase low baseline levels of NO and serotonin and decrease high baseline levels of testosterone and oxytocin in PE patients, with no effect on FSH, LH, prolactin and thyroid-stimulating hormone [105]. 

### 4.3. Topical Herbal Applications

Topical anesthetics, prominently lidocaine and prilocaine, have been widely used as a medical treatment to reduce glans sensitivity for PE [20]. In this review, four studies involved the application of the formula and washing off 30–60 min prior to intercourse [70,71,72,73]. The application of a Traditional Chinese Medicine Spray (TCMS) improved IELT and CIPE-5 intra-group but not significantly better than desensitization therapy (DT). However, the combination of TCMS and DT was significantly better than either of these interventions alone [70]. TCMS is composed of *Herba Asari*, *Syzygium aromaticum*, *Ootheca Mantidis*, *Radix Aconiti*, *Galla Chinensi*, *Fructus Rosae Laevigatae*, *Fructus Rubi* and *Pericarpium Zanthoxyli* [70]. The benefit of PE may be mediated through local anesthetic and analgesic effects (*Herba Asari*; *Pericarpium Zanthoxyli*; *Galla Chinensis*), relaxation of the corpus cavernosum (*Syzygium aromaticum*) and vasodilation (*Radix Aconiti*), although this has not been investigated [70]. Similarly, the application of the traditional Chinese medicine Secret Severance (SS) cream was shown to improve IELT in a dose-dependent manner compared to the control in two studies [71,72]. Furthermore, in a previous uncontrolled pilot study, SS-Cream improved IELT compared to baseline and seemed effective in PE and PE combined with erectile dysfunction [106]. SS-Cream is composed of *Ginseng Radix alba*, *Angelicae gigantis Radix*, *Cistanchis Herba*, *Torilidis Semen*, *Caryophylli Flos*, *Cinnamoni Cortex*, *Zanthoxyli Fructus*, *Asiasari Radix* and *Bufonis Veneum* [71,72]. Although the mechanisms of action of SS-Cream have yet to be determined, it is proposed that the active constituents of the included herbs act as desensitizing agents on the glans penis, smooth muscle relaxation and increased local blood flow that may be mediated through the activation of NO/guanosine monophosphate pathways [107]. However, the application of a Persian medicine polyherbal formula (*Nigella sativa* seed oil, *Olea europea* oil and oleo-gum-resin of *Boswellia sacra Flueck*) did not show any benefit intra-group or compared to placebo for IELT or PEDT [73]. 

### 4.4. Acupuncture and Electroacupuncture 

Acupuncture is a practice based on traditional Chinese medicine that involves inserting and manipulating needles into specific points of the body to manipulate meridians (energy channels) [108,109]. This meridian system is considered the core basis for therapies such as acupuncture and electroacupuncture, where acupuncture points (acupoints) on the skin guide “qi” energy from somatic meridians to internal organs [110]. In two studies, acupuncture increased IELT and decreased PEDT intra-group and compared to sham acupuncture in patients with PE. Although acupuncture was found to be as effective as paroxetine in one study [74], dapoxetine was found to be more effective than acupuncture in a separate study [76]. The application of electroacupuncture using the Zhongji point (−) and Sanyinjiao points (+) was found to improve IELT compared to baseline and more effectively than Longdan Xiegan [75]. Furthermore, electroacupuncture reduced serum testosterone significantly post treatment, as effectively as Longdan Xiegan [75]. However, the mechanisms by which acupuncture or electroacupuncture may be effective in the treatment of PE remain unknown and uninvestigated [74,75,76]. The insertion of needles or electric stimulation is proposed to activate local receptors, which deliver neuronal signals to the central nervous system that may mediate physiological activity [111]. At the acupoint level, this involves triphosphate (ATP) stimulation and transient receptor potential vanilloid (TRPV) channels. Centrally, acupuncture modulates neurotransmitters, including serotonin, dopamine, norepinephrine, opioids and endocannabinoids. Peripherally, acupuncture may reduce cyclooxygenase-2 (COX-2) and prostaglandin E2 (PGE2) [111,112]. Acupuncture has been shown to reduce pain and mediate analgesia, which is proposed through the modulation of various mechanisms such as increased serotonin centrally and peripherally, and the modulation of opioid peptides, enkephalin and dynorphin in the spinal cord [112,113]. NO may also mediate the benefit of acupuncture, where NO and NO synthase are increased around acupoints and meridian lines [114]. Alongside analgesia, acupuncture is beneficial for anxiety and depression through the modulation of serotonin [113].

### 4.5. Vitamin, Mineral and Nutraceutical Supplements

Vitamins, minerals and nutraceuticals were primarily investigated in just one study included in this review [77]. Furthermore, this study was the only study that investigated RE [77]. Sperm is recoverable in 69% of males with RE from post-orgasm urine samples for artificial reproductive techniques [6], alongside anticholinergics, sympathomimetics, surgery and electroejaculation [16]. In a crossover randomized open-label trial, amoxapine was found to increase the rate of antegrade ejaculation significantly higher compared to vitamin B12 [77]. Only 8% (n = 2) of 25 patients were improved due to vitamin B12, compared to 88% (n = 22) of patients taking amoxapine. Vitamin B12 is primarily involved in various physiological processes, including producing red blood cells and maintaining nerve function. Its deficiency can lead to anemia and neurological symptoms [115]. While retrograde ejaculation can be associated with certain neurological conditions or nerve damage, there is no established connection between RE and vitamin B12 deficiency. However, based on the information available, no scientific evidence suggests that vitamin B12 is used for the treatment of any EjD, specifically RE. 

### 4.6. Hyaluronic Acid 

Hyaluronic acid (also known as hyaluronan or hyaluronate) is a naturally occurring non-sulphated glycosaminoglycan found predominantly in the extracellular matrix that has diverse roles in human physiology, most notably tissue repair and regeneration through cell signaling and cell migration [116]. It is sourced from animal and bacterial sources for human health and disease [117] and used in bone and joint pathology, cartilage regeneration, ophthalmology, wound healing and dermatology [116]. 

In andrology, gland penis augmentation using hyaluronic acid has been applied to Peyronie’s disease and PE [118,119]. Although it is increasingly investigated for medical application, hyaluronic acid is not reported as a conventional treatment for PE [20] and is included as a TCAM treatment in this review. Of the included studies, just one investigated it as an intervention in a controlled clinical trial. The injection of hyaluronic acid into the glans penis and urethral meatus showed a significant improvement in IELT and AIPE following treatment compared to saline injection in a small study sample [78]. Previously, in an uncontrolled study, hyaluronic acid injection into the base of the penis increased IELT significantly from baseline scores [47]. This has been further supported in prospective studies, where hyaluronic acid may be safe and effective for PE and may improve satisfaction for up to 5 years [120,121,122,123]. Gland penis augmentation using hyaluronic acid is proposed to act as a bulking agent that blocks accessibility and inhibits tactile stimulation of the dorsal nerve receptors to improve ejaculatory control and ejaculation latency time [119]. 

### 4.7. Music Therapy

Music therapy is the application of personally tailored music interventions that involve music listening and/or music making. This is differentiated from music medicine, which does not include a personalized music intervention [124]. In this review, one study investigated music therapy as an intervention compared to dapoxetine. Using music therapy 45 min before sexual intercourse improved IELT and reduced PEDT compared to baseline and was as effective as dapoxetine [79]. Music therapy may particularly be valuable in stress and anxiety and for mental health benefits [124], which may explain the potential benefit of music therapy on PE. Furthermore, in obsessive-compulsive disorders, music therapy combined with SSRIs significantly improved obsessive compulsive disorder symptom scores alongside anxiety and depression scores compared to SSRI alone [125]. In adolescents and children with depression associated with attention deficit and hyperactivity disorder, music therapy increased serotonin secretion and reduced cortisol, blood pressure and heart rate [125]. In stressed rats, music therapy increased serotonin in the hippocampus and frontal cortex [126]. It may therefore increase serotonin and reduce anxiety in patients with PE. 

### 4.8. Strengths and Limitations 

This systematic review used the PRISMA guidelines to improve validity and reliability. The keyword search strategy and eligibility were based on the Population, Intervention, Comparison, Outcomes and Study Design (PICOS) framework and aligned with the aim of this study. As this review focused on a broad definition of TCAM, the interventions included for analysis are heterogeneous. The review identified a small number of studies included for EjD (n = 22), with the majority of these investigating PE. Therefore, the effects of TCAM in EjD remain unclear and under-investigated. The eligibility criteria did not limit to a specific outcome related to EjD. However, most of the included studies investigating PE used the IELT as a standardized measure, with or without a validated self-reported questionnaire, improving the validity of the included studies. The included studies focused on a cohort of patients with primary EjD diagnosis, with most studies on PE. However, there is some variation in the definition of the cohort for PE and inclusion criteria for the different studies. Therefore, the participants in the included PE studies were varied across different forms such as psychogenic organic, lifelong and/or acquired. Although English, Italian and Spanish full-text articles would be included, there is a limitation on other language publications. 

## 5. Conclusions

The results of this systematic review show that few controlled trials have investigated TCAM as an intervention in EjD. Although the majority of studies used IELT as a standardized outcome in PE, the available studies are widely heterogenous in the TCAM intervention investigated and comparison groups used. They mostly represent PE, and there is an underrepresentation of TCAM investigated in other types of EjD. In PE, most of the studies showed favorable outcomes intra-group and when compared to placebo. Based on these results, various TCAM interventions may have an important role particularly in the management of PE, while this remains unclear for other types of EjD. More standardized interventions need to be investigated in well-designed and controlled studies to confirm the benefit of specific TCAM interventions for EjD. 

## Figures and Tables

**Figure 1 medicina-59-01607-f001:**
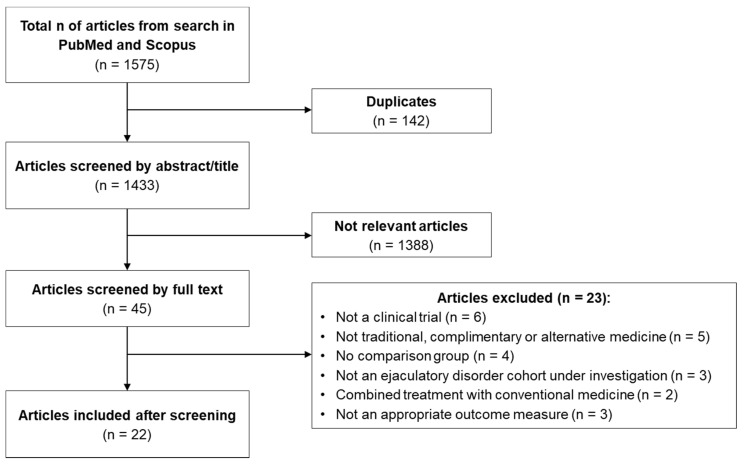
Flow diagram representing PRISMA search strategy.

**Figure 2 medicina-59-01607-f002:**
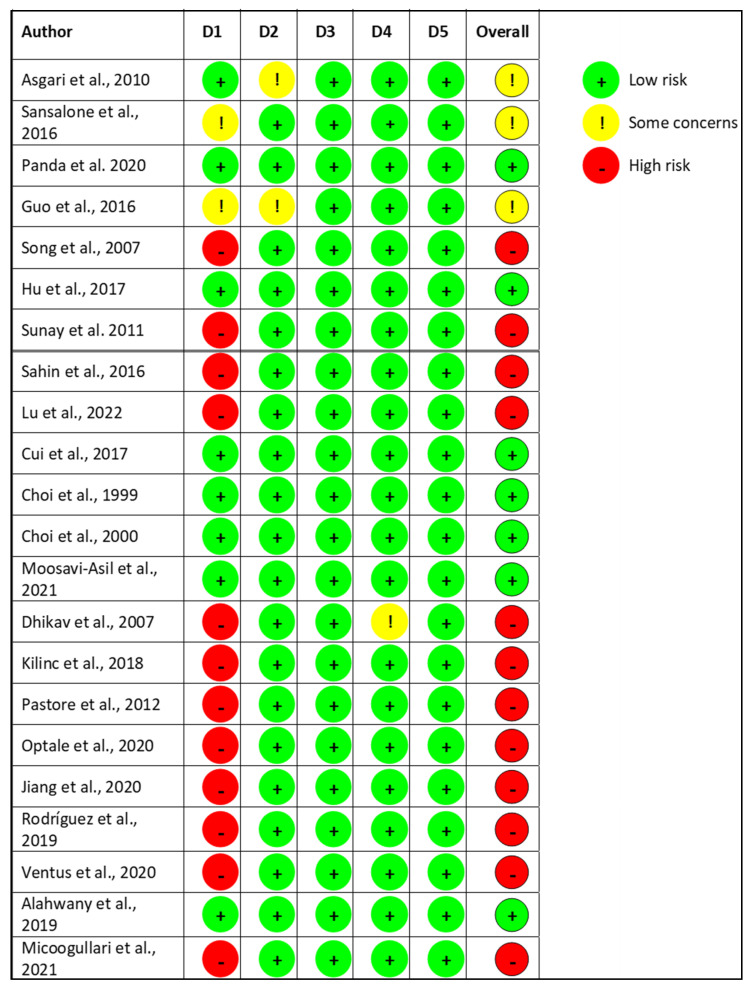
Cochrane Risk of Bias of the Included Studies. D1 = Randomization process; D2 = Deviations from the intended interventions; D3 = Missing outcome data; D4 = Measurement of the outcome; D5 = Selection of the reported result. Mentioned articles are cited in the bibliography as references [58,59,60,61,62,63,64,65,66,67,68,69,70,71,72,73,74,75,76,77,78,79].

**Table 1 medicina-59-01607-t001:** Characteristics of the studies included in this systematic review.

Ref.	Study Design	Patient Cohort	Intervention (n)	Comparison (n)	Outcomes of Interest	Other Outcomes	Adverse Effects
Lifestyle, Exercise and/or Physical Therapy							
Dhikav et al., 2007 [58]	Open-label control trial	PE patients	Yoga (2 pranayanams and 12 asnas) for 1 h daily for 12 weeks (n = 38)	Fluoxetine (20–60 mg) daily for 12 weeks (n = 30)	IELT increased significantly intra-group in both groups IELT increased inter-group for fluoxetine compared to yoga		Fluoxetine induced nausea, vomiting, anxiety and insomnia in some patients
Kilinc et al., 2018 [59]	RCT	LPE patients with IELT < 1 min and PEDT > 11	Moderate physical activity (running > 30 min 5 days weekly) for 30 days. (n = 35)	Sedentary lifestyle with dapoxetine (30 mg) on demand for 30 days (n = 35) and minimal physical activity (walking < 30 min 5 days per week) for 30 days (n = 35)	IELT increased significantly intra-group for all groups PEDT decreased significantly intra-group for moderate activity and dapoxetine groups IELT increased, and PEDT decreased significantly inter-group for moderate activity and dapoxetine compared to minimal activity IELT and PEDT were not significantly different for moderate exercise compared to dapoxetine		Yawning, nausea, dizziness and headache were reported for 5 dapoxetine patients.
Pastore et al., 2012 [60]	RCT	LPE with IELT < 1 min	PFM rehabilitation 60 min trice per week for 12 weeks (n = 19)	Dapoxetine 30 mg (n = 8) or 60 mg (n = 7) on demand for 12 weeks	IELT increased significantly intra-group for all groups, and it was higher for 30 mg and 60 mg of dapoxetine compared to PFM rehabilitation (although statistical analysis was not provided)		Nausea and diarrhea reported in 4 dapoxetine patients
Optale et al., 2020 [61]	RCT	LPE with IELT < 1 min	Mobile coaching application of physical and mental distancing exercises (10 min) 3 times a day for 3 months (n =17)	Verbal and printed instructions for physical and mental distancing exercises (10 min) 3 times a day for 3 months (n = 15)	PEDT decreased and PEP increased significantly intra-group for both groups Differences for PEDT and PEP were significantly higher for mobile coaching	IIEF-15 intercourse satisfaction increased significantly intra-group for mobile coaching IIEF-15 overall satisfaction increased significantly intra-group for mobile coaching Depression scores decreased significantly intra-group for both groups	Not reported
Jiang et al., 2020 [62]	RCT	Primary PE patients with IELT < 1 min for >1 year	PRM for 3 months (n = 18)	Kegel exercise for 3 months (n = 19)	IELT increased, and PEDT decreased significantly intra-group for both groups Differences for IELT and PEDT were significantly higher for PRM.		None
Rodríguez et al., 2019 [63]	Parallel RCT	PE patients with IELT of ≤2 min and PEDT of ≥11	GWD for 7 weeks (n = 18)	GWtD for 7 weeks (n = 17)	IELT increased significantly intra-group for both groups IELT increased significantly for GWD compared to the control		None
Ventus et al., 2020 [64]	RCT	Self-reported IELT < 3 min	VSS+ trice a week for 6 weeks. (n = 13)	VSS trice a week for 6 weeks (n = 12) and waiting list group control group (n = 5)	CHEES decreased significantly intra-group for both groups, and it was lower for VSS+	The Sexual Distress Scale, STAI and Brief Symptom Inventory-18 (Anxiety) scales decreased significantly intra-group for the VSS+ group.	None
Herbal Medicine Supplements							
Asgari et al., 2010 [65]	Double-blind RCT	PE patients with IELT < 2 min occurring >50% of coital attempts	*Hypericum perforatum* extract (150 mg containing 160 μg of hypericin) 3 times daily for 30 days (n = 22)	Placebo tablets 3 times daily for 30 days (n = 20)	IELT increased significantly intra-group for both groups IELT increased significantly for *Hypericum perforatum* compared to the control	IIEF-15 intercourse satisfaction, overall satisfaction and orgasmic function increased significantly intra-group for both groups and for *Hypericum perforatum* compared to the control IIEF-5 significantly decreased intra-group for both groups	Headache, constipation, photosensitivity and nausea were reported for *Hypericum perforatum* in 6 patients
Sansalone et al., 2016 [66]	Multi-center RCT	PE patients for ≥6 months and IELT of ≤2 min in >75% of coital attempts	EiacuMev^®^ (300 mg) daily for 3 months (n = 63)	Counselling and sexual behavioral treatment for 3 months (n = 65)	IELT increased, and PEDT decreased significantly intra-group for EiacuMev^®^ as well as compared to the control group MSHQ-EjD did not significantly change intra- or inter-group	FSH increased significantly intra-group for EiacuMev^®^ as well as compared to the control LH and Total Testosterone did not change significantly intra- or inter-group IIEF-5 increased significantly intra-group for EiacuMev^®^	Not reported
Panda et al., 2020 [67]	Single-center double-blind RCT	PE patients with IELT < 2 min and PEP ≥ 11	OLNP-05 (450 mg) twice daily for 8 weeks (n = 29)	Placebo capsule twice daily for 8 weeks (n = 28)	IELT and PEP increased significantly intra-group for OLNP-05, and compared to the control	CGI-I score decreased significantly intra-group for OLNP-05 as well as compared to the control	Constipation, fatigue and fever in 3 patients treated with OLNP-05 and in 5 controls
Guo et al., 2016 [68]	RCT	LPE > year with IELT ≤ 0.5 and ≤2 min and IIEF-5 > 21	QSF granules twice a day for 4 weeks (n = 29)	One bag of dapoxetine twice daily for 4 weeks (n = 30)	IELT increased significantly intra-group for both groups, and it was higher in the dapoxetine group compared to QSF PEDT decreased significantly intra-group for both groups, with no inter-group difference	CMSS decreased significantly intra-group for both groups, and it was lower in the QSF group Sex life satisfaction increased significantly intra-group for both groups, which was higher in the QSF group.	Slight discomfort in the stomach in 1 QSF patient; Dizziness and nausea in 5 controls
Song et al., 2007 [69]	RCT	PE patients with IELT < 2 min and partner satisfaction < 50%	Gu-jing-mai-si-ha (GJMSHT, Kanghabo Co., Ltd.) 4 tablets twice a day for 15 days (n = 35)	No drug (n = 33)	IELT, CIPE10 and CIPE5 scales increased significantly intra-group for GJMSHT as well as compared to the control	NO and PGF2α increased significantly intra-group for GJMSHT patients compared with the controls. Sexual Partner’s Satisfaction Rate and Wish Fulfillment Rate increased significantly intra-group for GJMSHT as well as compared to the control.	Not reported
Topical Herbal Applications							
Cui et al., 2017 [70]	RCT	PE patients with IELT < 2 min and CIPE-5 < 18	TCMS applied daily and 30 min before sexual intercourse and washed off before coitus for 6 weeks. (n = 29)	Desensitization Therapy (DT) 3 times per week for 6 weeks (n = 28) and combination of DT and TCMS for 6 weeks (n = 29)	IELT and CIPE-5 increased significantly intra-group for all groups, and it was higher for the combination group compared to TCMS and DT groups		Local burning sensation reported by some patients
Choi et al., 1999 [71]	Double-blind RCT	Primary PE patients only with IELT < 3 min and Sexual Satisfaction Rate < 50%	Application of SS-Cream (0.05, 0.1, 0.15 and 0.2 g) on glans penis 1 h before intercourse, washing off before sexual intercourse	Placebo cream on the glans penis 1 h before intercourse, washing off before sexual intercourse	IELT increased significantly in a dose-dependent manner for SS-Cream compared to the control Clinical efficacy based on IELT increased significantly in a dose-dependent manner for SS-Cream compared to the control.	Satisfaction Degree increased significantly in a dose-dependent manner for SS-Cream compared to the control. Clinical efficacy based on the Satisfaction Degree increased significantly in a dose-dependent manner for SS-Cream compared to the control.	Local mild burning sensation (14%)
			n = 50 patients in total taking each dose once for 250 test trials				
Choi et al., 2000 [72]	Double-blind RCT	LPE patients with ejaculatory latency < 3 min and/or Sexual Satisfaction Rate < 30%.	Application of SS-Cream (0.2 g) on glans penis 1 h before intercourse, washing off before sexual intercourse	Placebo (not specified) on glans penis 1 h before intercourse, washing off before sexual intercourse	IELT increased significantly for SS-Cream compared to the control Clinical efficacy based on IELT increased significantly for SS-Cream compared to the control.	Clinical efficacy based on sexual satisfaction increased significantly for SS-Cream compared to the control.	Mild local irritant symptoms including a sense of mild burning (n = 78), mild pain (n = 20), sporadic erectile dysfunction (n = 3), delayed ejaculation of more than 45 min (n = 4) and abrupt end of erectile response with failure to culminate with ejaculation (n = 5)
			n = 106 patients taking each dose 5 times for 530 test trials + 106 test trials with a placebo				
Moosavi-Asil et al., 2021 [73]	Triple-blind RCT	PE patients ≥ 6 months duration with IELT < 1 min, PEDT score > 11 and IIEF > 22	Application of Polyherb Formula (0.25 cc) on the penis glans and shaft each night and 1 h before intercourse for 3 weeks (n = 30)	Application of placebo base oil (0.25 cc) on the penis glans and shaft each night and 1 h before intercourse for 3 weeks (n = 33)	IELT and PEDT did not significantly change intra-group for both groups PE frequency and PE interpersonal difficulty improved significantly intra-group for the Polyherb Formula		Testicular pain (n = 1) and acute drug allergy (n = 1) that dropped out of the study
Acupuncture or Acupressure							
Sunay et al., 2011 [74]	RCT	PE patients with IELT of <2 min in >70% of coital attempts	Acupuncture twice per week for 4 weeks (n = 30)	Sham acupuncture twice per week for 4 weeks (n = 30) and 20 mg of paroxetine daily for 4 weeks (n = 30)	PEDT decreased significantly intra-group for acupuncture and paroxetine IELT increased, and PEDT decreased significantly for acupuncture and paroxetine compared to sham acupuncture		None
Sahin et al., 2016 [76]	RCT	Self-reported LPE patients	Acupuncture twice for 4 weeks (n = 29)	Sham acupuncture twice for 4 weeks (n = 28) and dapoxetine 30 mg (n = 28) or 60 mg (n = 27) for 4 weeks	IELT increased, and PEDT decreased significantly intra-group for all groups IELT increased significantly for dapoxetine 60 mg compared to acupuncture and for acupuncture compared to sham acupuncture PEDT was significantly lower for dapoxetine 60 mg, followed by dapoxetine 30 mg, acupuncture and sham acupuncture		In the dapoxetine groups, nausea, dizziness, diarrhea, insomnia and headaches were reported No side-effects were reported in the acupuncture groups.
Lu et al., 2022 [75]	RCT	Chinese medicine diagnosis for premature ejaculation	Electroacupuncture for 30 min once daily for 6 days, with 1 rest day, for 4 weeks (n = 25)	300 mL of Longdan Xiegan daily for 4 weeks (n = 25)	IELT increased significantly intra-group for both groups, and it was higher in electroacupuncture patients	Sexual Life Satisfaction Scores of Patients and Spouses increased significantly intra-group for both, and it was higher for electroacupuncture patients. Serum testosterone decreased significantly intra-group for both and was less for electroacupuncture patients.	Dizziness (n = 2) and subcutaneous hematoma (n = 3) in the electroacupuncture group Malady (n = 6), stomach discomfort (n = 4), mild diarrhea (n = 1) and dizziness (n = 1) in the controls
Vitamin, Mineral and/or Nutraceutical Supplements							
Hu et al., 2017 [77]	Open-label, randomized, crossover study	RE patients	Amoxapine (Pfizer, Japan) group, 50 mg daily for 4 weeks; After 1-week washout period, patients took 1500 µg of vit B12 for 4 weeks (n = 13)	Vit B12, 1500 µg for 4 weeks; After a 1-week washout period, controls took 50 mg of amoxapine daily for 4 weeks. (n = 12)	The success rate for the recovery of anterograde ejaculation was significantly higher for amoxapine compared to vit B-12		Amoxapine-treated patients reported sleepiness (n = 1) and constipation (n = 2); No adverse events Reported for vit B12
Hyaluronic Penile Injections							
Alahwany et al., 2019 [78]	Crossover RCT	PE patients with IELT < 1 min for LPE and <3 min	2 mL HA (25 mg/mL) into glans penis (6 injections at coronal level and 4 injections at urethral meatus) with evaluation at 1 week and 1 month; 18 months wash out, followed by crossover with saline injection (n = 15)	2 mL saline injection into glans penis (6 injections at coronal level and 4 injections at urethral meatus) with evaluation at 1 week and 1 month; 1 month wash out period, followed by crossover with HA injection (n = 15)	IELT and AIPE improved significantly after 1 month in comparison with the baseline and the control group		HA injection reported gradually decreasing local discomfort at the injection site (n = 3), glans penis ecchymosis (n = 2) and irregular blanched papule (n = 1). Saline injection reported injection site ecchymosis (n = 1). No adverse effects 1 month following injection.
Music Therapy							
Micoogullari et al., 2021 [79]	RCT	PE patients	Music therapy (the patient decided the type) for 45 min before the sexual intercourse for 60 days (n = 60)	Dapoxetine (30 mg) daily for 60 days (n = 60)	IELT increased, and PEDT decreased significantly intra-group in both groups IELT and PEDT were not significantly different in the music group compared to the dapoxetine group	STAI decreased significantly intra-group in both groups, and it was not significantly different between the groups	Not reported

Abbreviations: AIPE = Arabic Index of Premature Ejaculation; CGI-I = Clinical Global Impression-Improvement Scale; CHEES = Checklist for Early Ejaculation Symptoms; CIPE = Chinese Index of Sexual Function for Premature Ejaculation; CMSS = Chinese Medicine Symptoms Score; FSH = Follicular Stimulating Hormone; IELT = Intravaginal Ejaculatory Latency Time; IIEF = International Index of Erectile Function; LH = Luteinizing Hormone; LPE = Lifelong Premature Ejaculation; MSHQ-EjD = Men’s Sexual Health Questionnaire—Ejaculatory Disorder; NO = Nitric Oxide; PE = Premature Ejaculation; PEDT = Premature Ejaculation Diagnostic Tool; PEP = Premature Ejaculatory Profile; PGF2 = Prostaglandin F2; PRM = Penile Root Masturbation; RCT = Randomized Control Trial; RE = Retrograde Ejaculation; STAI = State-Trait Anxiety Inventory. Interventions: EiacuMev^®^ = *Satureya montana*, *Tribulus terrestris*, *Phyllanthus emblica*, Cardamomo, L-tryptophan, ascorbic acid, vitamins B1, B3 and B6; Gu-jing-mai-si-ha = *Radix anacycli pyrethri*, *Mastiche*, *Fructus cardamomi*, *Rhizoma Cyperi*, *Stigma Croci*, *Semen Myristicae*, *Radix Curcumae*, *Folium Syringae oblatae*, *Radix et Rhizoma Nardostachyos*, *Fructus Tsaoko* and *Flos Rosae rugosae*; GWD = Sphincter Control Training with Masturbation Device; GWtD = Sphincter Control Training without Masturbation Device; HA = Hyaluronic acid; OLNP-05 = *Mucuna pruriens*, *Cynara cardunculus*, *Trigonella foenum-graecum*, *Withania somnifera* and L-arginine; PFM = Physio-kinesitherapy, biofeedback therapy and electrical stimulation of perineal floor; Polyherb Formula = Nigella sativa seed oil, Olea europea oil and oleo-gum-resin of *Boswellia sacra Flueck*; QSF = Qiaoshao Formula—*Fructus Forsythiae* (20 g), *Radix Paeoniae Alba* (15 g), *Radix Bupleuri* (15 g), *Radix Astragali seu Hedysari* (10 g), *Morinda offcinalis How* (15 g), *Rhizoma Dioscoreae* (15 g) and *Rhizoma Acori Graminei* (5 g); SS-cream = *Ginseng Radix alba*, *Angelicae gigantis Radix*, *Cistanchis Herba*, *Torilidis Semen*, *Caryophylli Flos*, *Cinnamoni Cortex*, *Zanthoxyli Fructus*, *Asiasari Radix* and *Bufonis Veneum*; TCMS = Traditional Chinese Medicine Spray—*Herba Asari* (30 g), *Syzygium aromaticum* (30 g), *Ootheca Mantidis* (20 g), *Radix Aconiti*, (20 g), *Galla Chinensi* (20 g), *Fructus Rosae Laevigatae* (20 g), *Fructus Rubi* (20 g) and *Pericarpium Zanthoxyli* (1 g); VSS = Vibrator-assisted start–stop exercises; VSS+ = Vibrator-assisted start–stop exercises and mindfulness meditation.

## Data Availability

All data for the systematic review are provided in the manuscript (Table 1), and the source of the data is published in the public domain and available through references provided in the results.

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
