# Peer review of "Traditional, Complementary and Alternative Medicines in the Treatment of Ejaculatory Disorders: A Systematic Review"

_medicina, 2023, doi:10.3390/medicina59091607_

Round 1

Reviewer 1 Report

I didn't find inclusion criteria in articles collected.

it is suggested to add journals citation indices for each publication.

Author Response

Reviewer 1:

  1. I didn't find inclusion criteria in articles collected.

The methodology has been amended to clearly state the inclusion and exclusion criteria used for study eligibility (Ln 118 – 129): “The following inclusion criteria were used for eligibility of studies: i) a cohort of male patients diagnosed with any recognised EjD; ii) any sole TCAM intervention in at least one intervention group; iii) controlled clinical trials with any comparison group as study type; iv) any relevant outcome measures for EjD; v) English, Spanish, and Italian full text articles. The following exclusion criteria were used for eligibility of studies: i) no control group; ii) observational studies, review studies, case reports, conference proceedings, pre-prints, and any other grey literature.”

  1. It is suggested to add journals citation indices for each publication

Although publication citation indices can provide insight into research study impact and influence, they do not provide a direct measure of research quality, accuracy or relevance. Citation patterns can be confounded by numerous factors other than the merit or quality of the publication, and hence can be bias positively or negatively. Furthermore, journals citation indices for publications are not typically reported in PRISMA based systematic reviews. For the study results, we have complied with the PRISMA recommendations for results reporting applicable to systematic reviews. This includes study selection, study characteristics, study results, and risk of bias. For risk of bias, we used the Cochrane Risk of Bias tool, which is a widely used standard tool for a reflection of potential validity concerns in each publication. For these reasons, we would like to recommend that the citation indices for each publication do not provide further information that reflects the study characteristics, outcomes or quality.

Reviewer 2 Report

Dear authors

Thanks for the article. It's a nice review article. But there are some minor deficiencies and mistakes. You can see them in the attached pdf file.

Author Response

Reviewer 2:

  1. It would be more meaningful for the reader to add a paragraph about traditional knowledge medicine by making use of the following articles [Introduction: Ln 83]:
    • Ethnopharmacological study of medicinal plants in Kastamonu province (Turkiye)
    • Medicinal plants used in folk medicine of Akcaabat district (Turkey)
    • An ethnobotanical study of medicinal plants in Guce district, north eastern Turkey

An addition has been included in the introduction to capture traditional knowledge medicine using these recommended articles (Ln 87 – 90).

  1. The following articles should be added to Ln 310 and Ln 316 on original submission document:
    • Ethnopharmacological study of medicinal plants in Kastamonu province (Turkiye)
    • Medicinal plants used in folk medicine of Akcaabat district (Turkey)
    • An ethnobotanical study of medicinal plants in Guce district, north eastern Turkey

These articles have been included as citation on Ln 320 and Ln 325

  1. Re-Write referencing for Cooper et al. (2017) according to author guidelines (Ln 237-250)

This has been amended accordingly on Ln 248 – 258.

  1. Add full name for hypericum, - Hypericum perforatum L (Ln314)

This has been amended accordingly on Ln 323, 327 and 329

  1. Add full names for herbs – Ln 324

This has been amended accordingly on Ln 333 – 334.

Reviewer 3 Report

Compliments to the authors on a very interesting and exhaustive review on the traditional, complementary and alternative medicines for the management of Premature ejaculation.

A very well written and conducted study

Do you want to change your title and make it relevant only to premature Ejaculation? as it is there are no studies covering the other aspects of ejaculatory dysfunction (except one study) 

The therapy by hyaluronic acid is called glans augmentation. This should be mentioned in your discussion (Ref: Kosseifi F, et al. Glans penis augmentation using hyaluronic acid for the treatment of premature ejaculation: a narrative review. Transl Androl Urol. 2020 Dec;9(6):2814-2820.).

Few minor typo errors in the spelling:

Each (line 132),

thrice in table 1 page 6,

Author Response

Reviewer 3:

  1. Compliments to the authors on a very interesting and exhaustive review on the traditional, complementary and alternative medicines for the management of Premature ejaculation. A very well written and conducted study

The authors thank the reviewer for the positive comment and the recommendations provided.

  1. Do you want to change your title and make it relevant only to premature Ejaculation? as it is there are no studies covering the other aspects of ejaculatory dysfunction (except one study) 

The study was designed to review clinical application of TCAM in any form of ejaculatory dysfunction. As a systematic review was previously done specifically on CAM and premature ejaculation (Cooper at al., 2017), the aim was to broaden this into all forms of ejaculatory dysfunction and TCAM. Therefore, the inclusion of all forms of ejaculatory dysfunction was incorporated into the PICOS, which guided the keyword search criteria and the eligibility criteria for the inclusion of articles of interest. Following the pre-designed methodology, the results showed just 1 study in retrograde ejaculation, and no studies for other types of ejaculatory dysfunction. However, even though there are few or no papers in other forms of ejaculatory dysfunction, and the focus in the literature is on premature ejaculation as the most common form, we believe this should still be reported in line with the aim of the study.

We believe it is relevant to record in this review that there is an under-representation of TCAM clinical studies in other forms of ejaculatory dysfunction, and therefore have not changed the study to only focus on premature ejaculation. This is currently in the conclusion as ‘there is an underrepresentation of TCAM investigated in other types of EjD’. However, to further highlight this at the start of the discussion, we have added the following text (Ln 242 – 245): “These results further suggest that there is an under-representation of TCAM investigations in other clinically important forms of EjD, with only one study investigating the use of vitamin B12 in retrograde ejaculation, and no studies investigating other forms of EjD”.

  1. The therapy by hyaluronic acid is called glans augmentation. This should be mentioned in your discussion (Ref: Kosseifi F, et al. Glans penis augmentation using hyaluronic acid for the treatment of premature ejaculation: a narrative review. Transl Androl Urol. 2020 Dec;9(6):2814-2820.).

Thank you for the clarification provided in this recommendation. This has been added to the manuscript discussion in Section 4.6 on Ln 458 and Ln 468. The reference suggested has been used in the discussion.

  1. Few minor typo errors in the spelling: Each (line 132); thrice (table 1 page 6)

These errors have been amended accordingly 

We would like to again thank the reviewers for the recommendations provided.